# Machine Learning-Based Modeling of pH-Sensitive Silicon Nanowire (SiNW) for Ion Sensitive Field Effect Transistor (ISFET)

**DOI:** 10.3390/s24248091

**Published:** 2024-12-18

**Authors:** Nabil Ayadi, Ahmet Lale, Bekkay Hajji, Jérôme Launay, Pierre Temple-Boyer

**Affiliations:** 1Laboratory of Energy, Embedded System and Information Processing, National School of Applied Sciences, Mohammed First University, Oujda 60000, Morocco; ayadi.nabil@ump.ac.ma (N.A.); hajji.bekkay@gmail.com (B.H.); 2CNRS, LAAS, 7 Avenue du Colonel Roche, F-31400 Toulouse, France; ahmet_lale@hotmail.fr (A.L.); temple@laas.fr (P.T.-B.); 3INSAT, UT3-PS, INPT, University of Toulouse, 118 Route de Narbonne, CEDEX 9, F-31062 Toulouse, France

**Keywords:** silicon nanowire ion-sensitive field effect transistor SiNW-ISFET, extra trees regression (ETR), multi-layer perceptron (MLP), nonlinear regression (NLR), support vector regression (SVR), machine learning (ML)

## Abstract

The development of ion-sensitive field-effect transistor (ISFET) sensors based on silicon nanowires (SiNW) has recently seen significant progress, due to their many advantages such as compact size, low cost, robustness and real-time portability. However, little work has been done to predict the performance of SiNW-ISFET sensors. The present study focuses on predicting the performance of the silicon nanowire (SiNW)-based ISFET sensor using four machine learning techniques, namely multilayer perceptron (MLP), nonlinear regression (NLR), support vector regression (SVR) and extra tree regression (ETR). The proposed ML algorithms are trained and validated using experimental measurements of the SiNW-ISFET sensor. The results obtained show a better predictive ability of extra tree regression (ETR) compared to other techniques, with a low RMSE of 1 × 10^−3^ mA and an R^2^ value of 0.9999725. This prediction study corrects the problems associated with SiNW -ISFET sensors.

## 1. Introduction

Invented in 1970, the Ion-Sensitive Field-Effect-Transistor (ISFET) is described as a MOSFET device where the sensing layer is in direct contact with an analytic solution [1]. Thanks to its compatibility with the standard CMOS process [2,3], ISFET represents itself as a promising candidate in various applications in different fields, such as the food industry [4], the pharmaceutical industry [5,6], the biomedical field, environmental monitoring [7,8,9], agriculture applications [10], etc.

In spite of the interesting potentialities demonstrated by ISFET, there remain certain challenges limiting its performance, which must be addressed. These limitations include (but are not limited to) the ISFET sensitivity close to the theoretical limit, the Nernstian of 59 mV/pH and the response time, which is relatively high and not suitable for new applications requiring real-time detection.

To overcome these problems, researchers are exploring the potential of nanomaterials and nanotechnologies to develop small, low-cost, highly efficient and sensitive biosensors with a wide range of applications. Silicon nanowires (SiNW) and carbon nanotubes (CNT) are examples of promising sensing materials [11]. Their size, comparable to that of biological species, makes silicon nanowires (SiNW) an ideal interface between biological media and physicists’ tools. Moreover, their high surface-to-volume ratio makes them particularly sensitive to surface charge effects. These unique and unexpected properties of nanoscale materials make them ideal candidates for sensing applications.

The development of the ISFET sensor based on silicon nanowires (SiNW) appears to be the ideal candidate, offering good sensitivity thanks to a high surface-to-volume ratio [12,13,14,15]. Moreover, SiNW-ISFET sensors are inexpensive, easy to manufacture and reliable.

The sensing mechanism of SiNW-ISFET is based on the accumulation of charged molecules on the nanowire surface, which leads to a surface potential shift. The transistor then responds to changes in the surface potential with a threshold voltage shift ΔVth.

Metallic alloy nanoparticles possess excellent electronic properties, but their stability is limited due to low resistance to environmental factors, such as temperature, oxygen, and electromagnetic radiation. In contrast, oxide compounds, including complex iron oxides, provide greater stability at temperatures up to 1000 °C, making them promising for practical applications [16].

With the recent technological development of the Internet of Things (IoT), the SiNW-ISFET sensors can be also deployed at remote sites for the collection of data which will be transmitted in real time to the supervisor for analysis and decision making [17]. One of the fundamental monitoring parameters in these applications is the precise measurement of pH [18,19]. The development of accurate models for pH detection is essential to extract reliable information from the data collected.

Since 1992, several studies have focused on the modeling of the behavior of the ISFET sensor. Sergio Martinoia et al. were the first to develop a macro-model to simulate it under various physico-chemical conditions, including temperature variations [20]. This model is based on the site-binding model combined with the MOSFET transistor model. In addition, the ISFET model is developed and implemented in the SPICE simulation program. The detection behavior of Ta_2_O_5_ ISFET has been studied by Chuan et al., using Mathematica calculation software [21]. Andrzej Napieralski et al. modeled the behavior of the ISFET sensor using the mixed language VHDL-AMS [22]. B. Hajji et al. studied the influence of temperature on the pH-ISFET response using the Orcad PSPICE tool, and the Matlab R2006a software [23]. S. E. Naimi et al. presented the modeling of the pH-ChemFET response using the genetic algorithm [24]. The simplified version of the EKV model combined with the site binding model was used by S. E. Naimi et al. to describe the behavior of ISFET with respect to temperature and pH change.

A simplified model based on BSIM was proposed by N. Ayadi et al. to analyze the performance of the SiNW ISFET microsensor [25]. The use of the SiNW ISFET-based pH sensors for prolonged duration in various monitoring applications requires a robust ion-sensitive FET (ISFET) model to accurately predict pH values of unknown samples. The application of artificial intelligence (AI) to the SiNW ISFET sensor makes it possible to develop an efficient and precise model. This technique (ML) should be able to learn the patterns from labelled train data and apply these learning over unseen data for correct predictions. Four most popular supervised machine learning techniques are employed to learn the characteristics of the silicon nanowire ISFET sensor based on experimental data: neural networks or Multi-Layer Perceptron (MLP), Nonlinear Regression (NLR), Support Vector Regression (SVR) and Extra Trees Regression (ETR). The first works on modeling and compensation of ISFET temperature drift based on ML were presented by the authors of the papers [26,27,28,29]. These studies aimed to compare the performance of different ML models. Sinha et al. proposed the first ISFET temporal drift compensation model using an ML technique called Bayesian inference [29]. The use of these (ML) techniques can mitigate the need for complex subsystems reported in the literature for sensor drift compensation tasks. Nevertheless, no existing work has investigated the application of machine learning algorithms for modeling silicon nanowire (SiNW)-based ISFET sensor.

In this study, we focus on the development of an efficient model of the SiNW-ISFET sensor capable of efficiently predicting measurements of pH values using artificial intelligence (AI) techniques. Four models have been developed in this work and compared with real measurements in order to determine which one best guarantees the performance of the SiNW-ISFET sensor, namely: Multi-Layer Perceptron (MLP), Nonlinear Regression (NLR), Support Vector Regression (SVR), and Extra Trees Regression (ETR).

This paper is therefore organized in four sections: Section 1, introduction; Section 2, Modeling SiNW-ISFET using machine learning (ML) techniques; Section 3, results and discussion; and Section 4, conclusions.

## 2. Modeling SiNW-ISFET Using Machine Learning (ML) Techniques

The supervised learning used in this work is the machine learning (ML) task of learning a function that associates an input with an output based on examples of input–output pairs. Given a training set of n examples of input–output pairs (x1, y1), (x2, y2), … (xn, yn), machine learning (ML) generates each new value yj by the mapping function f defined by the following Equation (1). In this work, Y is associated with the current Ids.
(1)Y=f(X)

Through this equation, we deduce that ML models are trained using a set of labeled data (80% of the data), where the model learns each data type. Once the training process is complete, the model is tested on the test data (20% of the data) and then predicts the result.

All the algorithms used in this work are implemented using popular machine learning libraries in Python, such as scikit-learn and TensorFlow. These libraries offer a wide range of tools and features for efficiently creating, training and evaluating machine learning models.

In this work, we have tried the four most popular supervised machine learning techniques to learn the characteristics of the silicon nanowires ISFET sensor based on experimental data. These techniques are respectively, the Neural Networks or the Multi-Layer Perceptron (MLP), the Nonlinear Regression (NLR), the Support Vector Regression (SVR) and Extra Trees Regression (ETR) (Figure 1).

The aim of examining these different ML algorithms is to select the best algorithm that seems closest to the assigned function, and to choose the right combination of hyperparameters.

### 2.1. Multi-Layer Perceptron Neural Networks

The Artificial Neural Networks are powerful tools for the simulation and prediction of I-V characteristics of ISFETs sensors. In particular, Multilayer Perception (MLP) is considered as the most common Neural Network model (Figure 2), consisting of successive linear transformations followed by processing with non-linear activation functions.

In this work, the developed MLP is a feed forward network including three layers as shown in Figure 2:
The input layer with five neurons; each neuron is associated with a parameter such as the Vgs, the nanowire length, the number of wires, the gate lengths and the pH;One hidden layer of Multilayer Perception;An output layer with a single neuron associated with the courant Ids.

A split ratio of 80–20% is used to divide the data into training and testing sets. MLP is trained with 80% experimental data using the Levenberg Marquardt (LM) algorithm for faster convergence, and 20% experimental data are used for the test.

The output of MLP Ids is determined as a nonlinearity activation function, applied to the linear combination of the input vector presented by Equation (1) [30,31].
(2)$$I_{ds}=Y=\sum_{i=1}^{s}\gamma_{ij}w_{ij}+b_{j}\mathrm{and}\;j=1,2,\ldots n$$
where Wij is the weight connecting the jth neuron in the output layer; b is the bias termand; and n is the number of neurons in the output layer.

The output activation $\gamma_{lj}$ is given by:(3)$$\gamma_{lj}=f(n_{lj})$$

The hidden and the output layers are processed using a softmax activation function as shown in Equation (4):(4)$$\gamma_{lj}=\frac{e^{n_{lj}}}{\sum_{k=1}^{n}e^{n_{kj}}}$$
$n_{lj}$ is the net input given by:
(5)$$n_{lj}=\sum_{i=1}^{q}w_{il}x_{ij}+b_{l}\mathrm{and}l=1,2\ldots ,s$$
where s is the number of neurons in the hidden layer; q is the number of neurons in the input layer; b_l_ is the bias term; and w is the weighting factor [31]. In this work, the main MLP key parameters (Equation (5)) are defined as follows:
$w_{ij}$ is the connecting weight;$x_{i}^{\prime }(i=1,\ldots ,10)$ is the output from ith neuron of the first hidden layer;$I_{ds}(MLP)$ is the output of the only neuron in the output layer.


The input layer includes five neurons; each neuron is associated with a parameter such as the V_gs_, the nanowire length ($l_{nw}$), the number of wires (N), the gate lengths (L) and the pH;
(6)$$\left(\begin{array}{c} \begin{array}{c} x_{11}\\ x_{21}\\ x_{31} \end{array}\\ x_{41}\\ x_{51} \end{array}\right)=\left(\begin{array}{c} \begin{array}{c} V_{gs1}\\ l_{nw1}\\ pH1 \end{array}\\ L1\\ N1 \end{array}\right)$$

For the first hidden layer, we replace the input layer in Equation (5):(7)$$\left(\begin{array}{c} n_{11}\\ \vdots \\ n_{101} \end{array}\right)=\left(\begin{array}{ccc} w_{11} & \cdots & w_{51}\\ \vdots & \ddots & \vdots \\ w_{110} & \cdots & w_{510} \end{array}\right)\left(\begin{array}{c} \begin{array}{c} V_{gs1}\\ l_{nw1}\\ pH1 \end{array}\\ L1\\ N1 \end{array}\right)+\left(\begin{array}{c} b_{1}\\ \vdots \\ b_{10} \end{array}\right)$$

The first hidden layer uses a softmax activation function, therefore, it is noted as:(8)$$\left(\begin{array}{c} x_{11}^{\prime }\\ \vdots \\ x_{101}^{\prime } \end{array}\right)=\left(\begin{array}{c} \frac{e^{n_{11}}}{e^{n_{11}}+\ldots +e^{n_{101}}}\\ \vdots \\ \frac{e^{n_{101}}}{e^{n_{11}}+\ldots +e^{n_{101}}} \end{array}\right)$$

For the output layer, replacing Equation (8) in Equation (2), the new equation is written as follow:(9)$$I_{ds1}(MLP)=w_{11}x_{11}^{\prime }+w_{21}x_{21}^{\prime }+\ldots +w_{101}x_{101}^{\prime }+b_{1}$$

Finally, to determine the model output, the flowing set up is processed, as noted in the Program flow chart of Multilayer Perception (MLP) (Figure 3).
Calculation of the Error
(10)$$e=I_{ds}\left(exp\right)-I_{ds}\left(MLP\right)$$


Weight Updating(11)$$\left(\begin{array}{c} \begin{array}{c} ∆w_{11}\\ ∆w_{21}\\ \vdots \end{array}\\ ∆w_{101} \end{array}\right)=\alpha e\left(\begin{array}{c} \begin{array}{c} w_{11}\\ w_{21}\\ \vdots \end{array}\\ w_{101} \end{array}\right)$$
where α is the learning rate.



New Weights Calculation

(12)
$$\left(\begin{array}{c} \begin{array}{c} w_{11}(new)\\ w_{21}(new)\\ \vdots \end{array}\\ w_{101}(new) \end{array}\right)=\left(\begin{array}{c} \begin{array}{c} w_{11}\\ w_{21}\\ \vdots \end{array}\\ w_{101} \end{array}\right)+\left(\begin{array}{c} \begin{array}{c} ∆w_{11}\\ ∆w_{21}\\ \vdots \end{array}\\ ∆w_{101} \end{array}\right)$$



Figure 3 illustrates the functioning of the Multilayer Perceptron (MLP) algorithm. The process begins with downloading the dataset, followed by preparing the input and output variables. Next, the input features are normalized to ensure a uniform scale across the dataset. The model is then defined and trained on 80% of the data, using the backpropagation gradient algorithm for training. Evaluation is based on calculating the error e (the difference between Ids calculated and Ids measured), which should be minimized to assess the model’s performance across multiple subsets of the dataset. Once trained, the models are deployed to predict new data instances. Finally, the error indicators R^2^, RMSE, and MAE are used to effectively evaluate the predictive capability of the model.

### 2.2. Support Vector Regression

Support Vector Machines (SVM) are supervised learning models with associated learning algorithms that analyze data used for classification and regression analysis. They are very effective at analyzing input and output parameters because of their global minima. The SVM method is called support vector regression (SVR), when used for prediction (Figure 4). In fact, it represents a nonlinear regression algorithm where the data sample inputs are represented in a high-dimensional feature domain for nonlinear matching [32,33,34].

The use of the SVR method requires a set of training data determined by {(xi,yi),i=1,2,...,n} where xi is the ith element of input vector in n dimensional space and yi is the actual value corresponding to xi. In the case study, the input vectors are respectively the gate voltage (V_gs_), the nanowire length ($l_{nw}$), the number of wires (N), the gate lengths (L) and the (pH). While, the output is the source drain current ($I_{ds}$). Furthermore, the evaluation of the main key parameters is based on the estimation function f(x) shown in Equation (13) [34]:(13)$$y=f(x)=w^{T}\varphi (x)+b$$
where f(x) denotes the forecasting values; φ(x) is the mapping function of kernel space for extracting the character from the original space; w is a weight vector; and b is the bias.

To minimize the structure risk and reduce the model complexity with respect to the training data, a sensitive loss function ε is integrated in the SVR as suggested by Vabnik [35]. Obviously, it is used when the error is not penalized as long as it is less than ε. This function is able to transform the regression problem into an optimization one as described by Equation (14):(14)$$L_{\varepsilon }=\left\{\begin{array}{c} 0\;\;\;if\;\left|y-f(x)\right|\leq \varepsilon \\ \left|y-f(x)\right|-\varepsilon \;\;\;\mathrm{otherwise} \end{array}\right.$$

For more details, the SVR method objective is to find the optimum hyperplane and minimize the error between the training data and ε-insensitive loss function. This could be performed by minimizing the overall errors, as follows [36]:(15)$$min:\frac{1}{2}w^{T}w+C\sum_{i=1}^{N}\max(0,|y_{i}-w^{T}\varphi \left(x_{i}\right)-b|-\varepsilon )$$
where C is the penalty applied over training errors and ε is the user-specified positive constant.

#### Kernel Function

The Kernel function is one of the main of SVR function. In fact, the SVR performance is deeply dependent on the selected kernel function parameters. In this work, the Gaussian kernel (RBF Kernel) function G(xi, xj) has been employed to get the best performance and adjust the Key Kernel parameter (γ) [37]:(16)$$G\left(xi,xj\right)=\exp(-\gamma {\left\|x_{i}-x_{j}\right\|}^{2}),\gamma >0$$
where γ is the Kernel parameter.

Finally, the Support Vector Regression (SVR) function (f(x)) for predicting the new values can be formulated as:(17)$$f\left(x\right)=\sum_{i=1}^{N}\left(\alpha_{i}-\alpha_{i}^{*}\right)G\left(x_{i},xj\right)+b$$
where $\alpha_{i}$ and $\alpha_{i}^{*}$ are the Lagrange multipliers (i.e., a single weight) subject to $\forall i:0\leq \alpha_{i},\alpha_{i}^{*}$ and $\forall i:\alpha_{i}\alpha_{i}^{*}=0$.

Support Vector Regression (SVR) is described using the following steps:

Step 1: State the training set {(xi,yi),i=1,2,...,n}


Step 2: Select the appropriate parameters C, γ, ε, where C > 0, γ>0 and ε > 0

Step 3: Solving problem (15) and we get the solution ($w^{T}$, b)

Step 4: Select the feature set: $\alpha_{i}$ and $\alpha_{i}^{*}$


Step 5: Construct the decision formula (17): $f\left(x\right)=\sum_{i=1}^{N}\left(\alpha_{i}-\alpha_{i}^{*}\right)G\left(x_{i},xj\right)+b$


Figure 5 represents the program flow chart of the Support Vector Regression (SVR) with the set of resolution steps. Initially, the data are integrated as a form of experimental input (the gate voltage (Vgs), the nanowire length (lnw), the number of wires (N), the gate lengths (L) and the (pH)) and output (Ids experimental). Thereafter, the choice of the Kernel function and relaxation factors *C* and γ is made. After that, the data training and testing are performed. Finally, the source-drain current (Ids) predicted is assessed.

### 2.3. Nonlinear Regression

Nonlinear regression (NLR) is a technique of regression analysis in which observational data are controlled by a specified function. This one is a nonlinear combination of model parameters that depends on one or more independent variables. In fact, the data are adapted by a method based on successive approximations. These data are used in similar cases without additional training [38].

The use of the NLR method requires a set of training data determined by {(xi,yi),i=1,2,...,n} with a size value of n. Moreover, the explicit data-driven model can be written as follows [39]:(18)y=f(x,β)+ε
where y∈IR is the response variable, $x=(x_{1},...,x_{k})\in {IR}_{k}$ are the explanatory variables and $\beta =(\beta_{1},...,\beta_{p})\in {IR}_{p}$ are the estimated parameters. Indeed, the f represents the regression function, whose form is known up to some unknown parameters β, and error term ε with zero mean and variance $\sigma^{2}$. Following the approach of the linear model, if β is a vector of length d, there are n−d degrees of freedom. Then, the residual error variance is defined as:(19)$$\sigma^{2}=\frac{1}{n-d}\sum_{i=1}^{n}{\left(y_{i}-f\left(x_{i},\beta \right)\right)}^{2}$$

The unknown parameter vector β in the nonlinear regression model is estimated from the data by minimizing the residual sum of square (Equation (20)) [40]. This estimation is also known as nonlinear least squares:(20)$$S(\beta )=\sum_{i=1}^{n}{\left(y_{i}-f\left(x_{i},\beta \right)\right)}^{2}$$

A normal distribution for the error term is given by:(21)$$y-f\left(x,\beta \right)=\varepsilon \sim N(0,\sigma^{2})$$

If the errors εi follow a normal distribution N (0, σ^2^), then the least squares estimator for β is also the maximum likelihood estimator. Therefore, the likelihood for the nonlinear regression model is expressed as [41]:(22)$$L\left(\beta ,\sigma^{2}\right)=\frac{1}{{\left(2\pi \sigma^{2}\right)}^{\frac{n}{2}}}\exp(-\frac{S(\beta )}{2\sigma^{2}})$$

In this work, the researched parameters are the source drain current (Ids) (output) estimated as a function of the gate voltage (Vgs), the nanowire length (lnw), the number of wires (N), the gate lengths (L) and the (pH) (inputs). From the different formulations for the (NLR) function for the prediction of data, the following formula is developed:(23)$$\begin{array}{cc} y= & \frac{x_{5}\beta_{1}\beta_{2}\beta_{5}}{1+(\beta_{3}+\beta_{1}\beta_{4}\left(x_{2}-x_{3}\right))(x_{1}-\beta_{6}-\beta_{7}x_{4})}[(\log(1\\ & {+\exp\left(\frac{x_{1}-\beta_{6}-\beta_{7}x_{4}}{\beta_{8}\beta_{2}}\right)))}^{2}\\ & -{\left(\log\left(1+\exp\left(\frac{x_{1}-\beta_{6}-\beta_{7}x_{4}-\beta_{2}}{\beta_{8}\beta_{2}}\right)\right)\right)}^{2}]+\varepsilon \end{array}$$

The main (NLR) key parameters shown in Equation (25) are defined as follows: Ids (NLR) is the source-drain current (output symbolized by y); the inputs (x1, x2, x3, x4, x5) are respectively the gate voltage Vgs, the nanowire length (lnw), the gate lengths (L), the pH and the number of wires (N):(24)$$\left(\begin{array}{c} \begin{array}{c} x_{1}\\ x_{2}\\ x_{3} \end{array}\\ x_{4}\\ x_{5} \end{array}\right)=\left(\begin{array}{c} \begin{array}{c} V_{gs}\\ l_{nw}\\ L \end{array}\\ \mathrm{pH}\\ N \end{array}\right)$$

The model output is determined, replacing Equation (24) in Equation (23). Then, the new equation can be written as follows:(25)$$\begin{array}{cc} I_{ds}(NLR)= & \frac{N\beta_{1}\beta_{2}\beta_{5}}{1+(\beta_{3}+\beta_{1}\beta_{4}\left(l_{nw}-L\right))(V_{gs}-\beta_{6}-\beta_{7}pH)}[(\log(1\\ & +\;{\exp\left(\frac{V_{gs}-\beta_{6}-\beta_{7}pH}{\beta_{8}\beta_{2}}\right)))}^{2}\\ & -{\left(\log\left(1+\exp\left(\frac{V_{gs}-\beta_{6}-\beta_{7}pH-\beta_{2}}{\beta_{8}\beta_{2}}\right)\right)\right)}^{2}]+\varepsilon \end{array}$$

The developed model set up for the output prediction is also presented in the program flow chart of nonlinear regression (NLR) (Figure 6). For the model convergence, the residual error variance (σ^2^) is calculated using the following equation:(26)$$\sigma^{2}=\frac{1}{n-d}\sum_{i=1}^{n}{\left({Ids}_{i}(exp)-{Ids}_{i}(NLR)\right)}^{2}$$

Figure 6 shows the program flow chart of the Nonlinear Regression (NLR) with the set of resolution steps. Initially, the data are included as forms of experimental inputs (the gate voltage (Vgs), the nanowire length (lnw), the number of wires (N), the gate lengths (L) and the (pH)) and output (Ids experimental). After that, the (NLR) function, the initial coefficients for the nonlinear model β_0_, are chosen. Next, the data training and testing operation are carried out. Finally, the predicted source-drain current (Ids) is evaluated.

### 2.4. Extra Trees Regression

Extra Trees Regression is an ensemble learning method designed for regression tasks. This algorithm, an extension of the Random Forest Regression model, was proposed by Geurts et al. [42]. Like Random Forest, Extra Trees constructs a collection of decision trees, but it incorporates additional randomness during the training process. Extra Trees are particularly important within this class of algorithms and have demonstrated state-of-the-art performance on various regression tasks, especially with high-dimensional inputs and outputs [43]. They are computationally efficient, offering faster training and working effectively with both categorical and numerical data. Additionally, Extra Trees can manage non-linear relationships between features and target variables.

Random Forest (RF) is a regression technique that enhances predictive accuracy by combining the outputs of multiple decision tree (DT) algorithms to classify or predict the value of a variable [44,45]. When RF receives an input vector (x), consisting of the values of various features evaluated for a specific training set, it constructs K regression trees and averages their outputs. After growing K trees ${\left\{T(x)\right\}}_{1}^{K}$, the RF regression predictor is computed as the average of the individual tree predictions, based on the equation:(27)$${\hat{f}}_{RF}^{K}(x)=\frac{1}{K}\sum_{k=1}^{K}T(x)$$

The Extra Trees Regression (ETR) and Random Forest (RF) models differ in two significant ways. First, ETR considers all potential split points and randomly selects from them to divide nodes. Second, it utilizes the entire training dataset to grow the trees, helping to minimize bias [42].

In the ETR model, the splitting process is controlled by two key parameters: k and n_min_. k defines the number of features randomly chosen at each node, while n_min_ specifies the minimum number of samples required to split a node. These parameters enhance accuracy and reduce the risk of overfitting in the ETR model [43,46].

In our study, we trained an Extra Trees regression model with 100 trees to predict the source-drain current (Ids) of the SiNW-ISFET sensor. This approach was implemented using the Python programming language. The developed model for output prediction is also illustrated in the program flowchart for Extra Trees regression (ETR) (Figure 7).

### 2.5. Error Indicators

In this work, four performance indices, including the coefficient of determination (R^2^), the Mean Absolute Error (MAE) and the Root Mean Square Error (RMSE), were used to assess the accuracy of the forecast results (simulated results). These indices are generally used to evaluate the accuracy of the developed models in comparison with experimental results. The following Table 1 indicates the indices formulas used in this work.

## 3. Results and Discussion

In this section, we present the results obtained for the four artificial intelligence methods used for the learning of the silicon nanowire ISFET sensor, namely neural networks or Multi-Layer Perceptron (MLP), Nonlinear Regression (NLR), Support Vector Regression (SVR) and Extra Trees Regression (ETR).

For the machine learning techniques developed, around 956 datasets were used in the simulation of the SiNW-ISFET sensor in MATLAB R2018a software and the Python language. The minimum and maximum data ranges used for these models are shown in Table 2.

For the experimental activities, the Laboratory of Analysis and Architecture of Systems (LAAS-CNRS, Toulouse, France) provided a set of data obtained with N+/P/N+ silicon-nanowire-based ion-sensitive field effect transistors SiNW-ISFET [15]. These devices were realized using silicon technologies thanks to a specific reactive ion etching process for the silicon nanowire fabrication, and were finally adapted to liquid phase analysis thanks to the integration of an SU-8 based microfluidic channel by wafer-level packaging (to be published). The Al_2_O_3_ detection layer was deposited by ALD. The thickness of this layer is about 3.3 nm for 30 ALD cycles, and about 4.5 nm for 50 ALD cycles. The composition of this layer at the interface evolves very gradually from silicon to alumina Al_2_O_3_; we will call it aluminosilicate AlxSiyOz. The alumina beyond the interface zone is stoichiometric; after calibration of the EDX with two samples, one in sapphire Al_2_O_3_ and the other in SiO_2_, the ratio between oxygen and aluminum is indeed 60:40 [48].

It has been shown that the refractive index of alumina deposited by ALD increases under the effect of heat treatment. This phenomenon could be explained by crystallization of the films. To demonstrate this, the alumina was characterized using grazing incidence X-ray diffraction (GIXRD). This is a non-destructive technique used to study crystalline structures. The sample is subjected to an X-ray with a wavelength close to the dimensions of the crystal lattice (around 0.1 nm). The crystalline planes then act as a lattice of slits whose spacing is close to the wavelength of the X-rays. Diffraction then occurs at a precise angle that depends on the crystalline structure of the sample being analyzed. If the structure is amorphous, there is no crystalline plane and therefore no diffraction [49].

In the case of the developed ML models, the dataset comprising 956 samples was partitioned, with approximately 80% (765 samples) used for training and 20% (191 samples) reserved for testing.

Figure 8 illustrates the comparison between the experimental measurements of the SiNW-ISFET sensor (line) and the simulation results for the four machine learning techniques. The lengths of wire used are respectively 20 µm and 4 µm and the drain-source voltage is fixed at 1 (Vds = 1 V).

From the results, the transfer characteristics Ids(Vgs) show good scalability as a function of the different lengths of nanowire. Obviously, the effect of the wire length is due to parasitic contact resistance (RS1 + RS2). As the wire length increases, the transconductance decreases. A pronounced saturation phenomenon also occurs as the nanowire length increases, leading to a decrease in the on-state current of the transistor. From this figure, a good agreement between the (ML) models and the measurements is observed. Furthermore, the Extra Trees Regression (ETR) technique shows a very good prediction (as shown in Figure 9) of the SiNW-ISFET characteristics, with a coefficient of determination (R^2^) of 0.99998 and root mean square error of (RMSE) of 0.0027 µA.

The effect of different numbers of nanowires (1, 20 and 100) on the Si-NW-ISFET performance has been studied and presented in Figure 10. The results indicate a monotonic variation in the Ids current with the number of nanowires. This suggests that the series resistance R_S_ is negligible compared to the on-resistance R_ON_ of the nanowire, as shown by the proportionality of the transconductance with the number of nanowires. From the comparison of the four models, the ML model give accurate results, in good agreement with the experiments. In particular, the Extra Trees Regression (ETR) model proves its effectiveness and good prediction (as shown in Figure 11) for the assessment of the SiNW-ISFET characteristics (R^2^ of 0.99997 and RMSE of 0.002 mA).

Figure 12 shows a decrease in the drain current of the SiNW-ISFET sensor with an increase of the channel length L (short channel: 0.73 μm and long channel: 3.73 μm) for a fixed source-drain voltage (Vds = 1 V). In fact, this current decreases with the W/L ratio increase in the transistor ON state. This phenomenon is related to the length of the wire. Since the current in the saturation region is theoretically proportional to W/L, this measurement result is consistent with the theoretical prediction.

For the test accuracy of the four developed models, Figure 13 illustrates the varying efficiency of the compared machine learning (ML) techniques. As results, the Extra Trees Regression (ETR) technique seems to conform to the measurement, and gives very accurate results and a correct forecast (as shown in Figure 13) (R^2^ of 0.99996 and RMSE of 0.0015 mA).

In Figure 14, the transfer characteristics Ids (Vds) of SiNW-ISFET with varying pH values are presented. The measured data are described by line and the simulations results with other symbols for the four machine learning techniques. From this figure, it is confirmed that the surface potential at the Al_2_O_3_/SiO_2_/SiNW interface is well modulated by the pH values. The average shift of the threshold voltage Vth by pH was 51.66 mV/pH (Figure 14); in comparison with the Nernst ideal limit (∼59.2 mV/pH at 25 °C).

The results of Figure 15 prove that the ML models match perfectly with the measured data. A very low error is noted for the Extra Trees Regression (ETR) model with an R^2^ and RMSE about (0.99998, 0.53 µA). For the sensibility (Figure 16), it is noticed that Extra Trees Regression (ETR) provides an accurate sensibility in comparison with the others models.

For more effective analysis of the four machine learning techniques, a three of uncertainty indices MAE, RMSE and R^2^ have been reported in Table 3. Moreover, the parameters used for the development of the ML models are presented in Table 4. Indeed, it is clear that the results obtained from the four machine learning algorithms are very promising. Furthermore, the Extra Trees Regression (ETR) model allows very accurate results and correct prediction with a mean R² of 0.9999725, MAE of 0.000415 mA and RMSE of 0.001 mA.

## 4. Conclusions

In this work, four mathematical models were developed for the silicon nanowire ISFET sensor using MATLAB software and the Python language: Multi-Layer Perceptron (MLP), Nonlinear Regression (NLR), Support Vector Regression (SVR), and Extra Trees Regression (ETR). The objective was to create highly accurate predictive models for the source-drain current (Ids) output of the SiNW-ISFET sensor. The four developed models have been compared with real measurements in order to determine the one that ensures the best performance. From the results, the ML models are found to be efficient. In particular, Extra Trees Regression (ETR) offers the best performance prediction. It presents a good agreement with the experimental results with a lowest RMSE value of about 0.001 mA and R^2^ of 0.9999725.

This model proves to be effective for studying the simulation response of the silicon nanowire ISFET and analyzing various parameters related to its manufacturing process. The use of artificial intelligence techniques is of great importance, as it allows for the prediction and correction of defects before manufacturing. Thereafter, the application of these machine learning methods could be very promising for future studies of CMOS circuits.

## Figures and Tables

**Figure 1 sensors-24-08091-f001:**
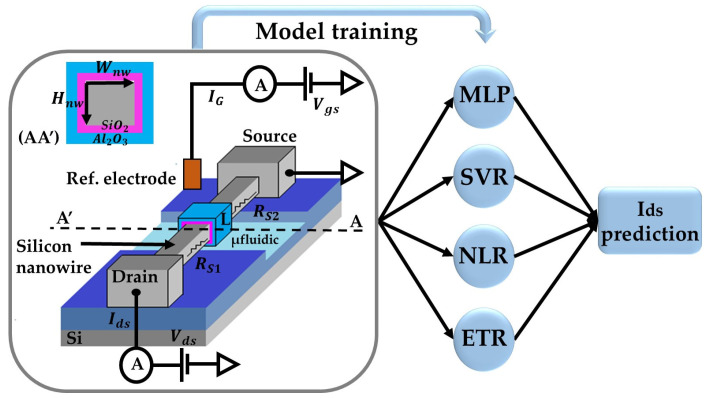
The (ML) techniques used for the Ids prediction of the silicon nanowires ISFET.

**Figure 2 sensors-24-08091-f002:**
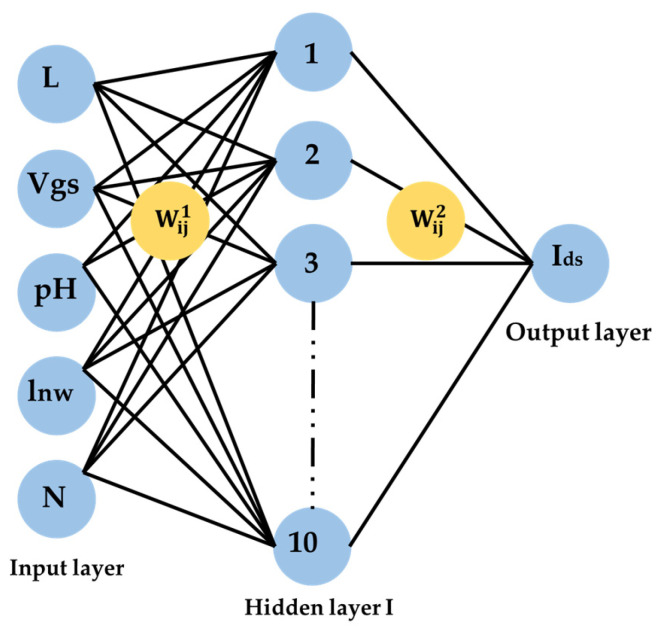
The Multi-Layer Perception Neural Networks (MLP).

**Figure 3 sensors-24-08091-f003:**
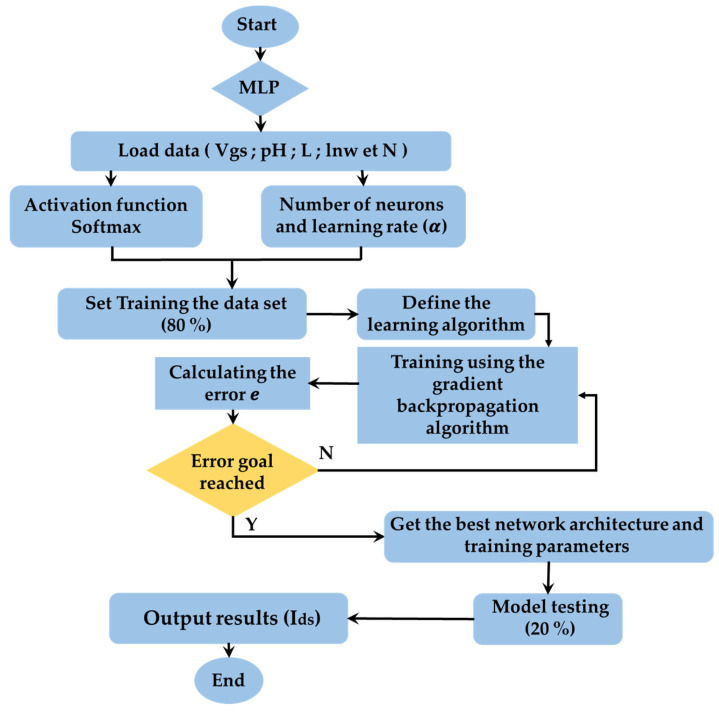
Program flow chart of Multilayer Perception (MLP).

**Figure 4 sensors-24-08091-f004:**
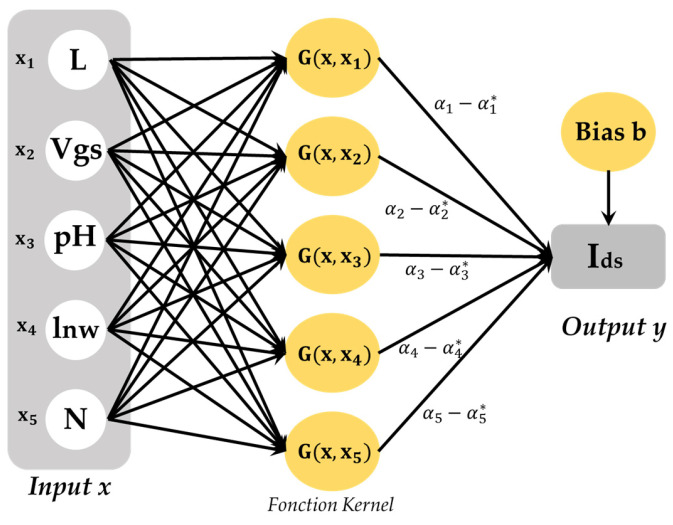
Architecture of Support Vector Regression (SVR).

**Figure 5 sensors-24-08091-f005:**
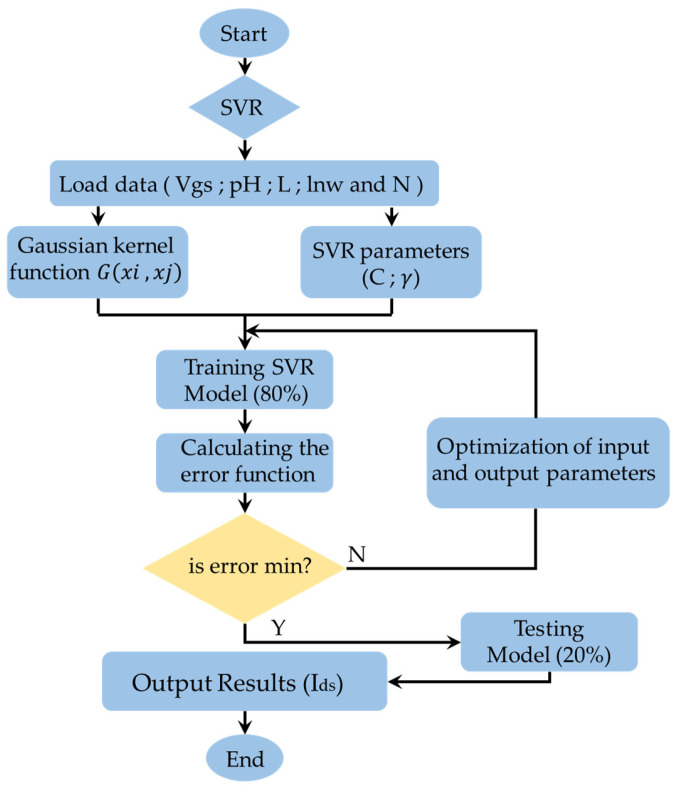
Program flow chart of the Support Vector Regression (SVR).

**Figure 6 sensors-24-08091-f006:**
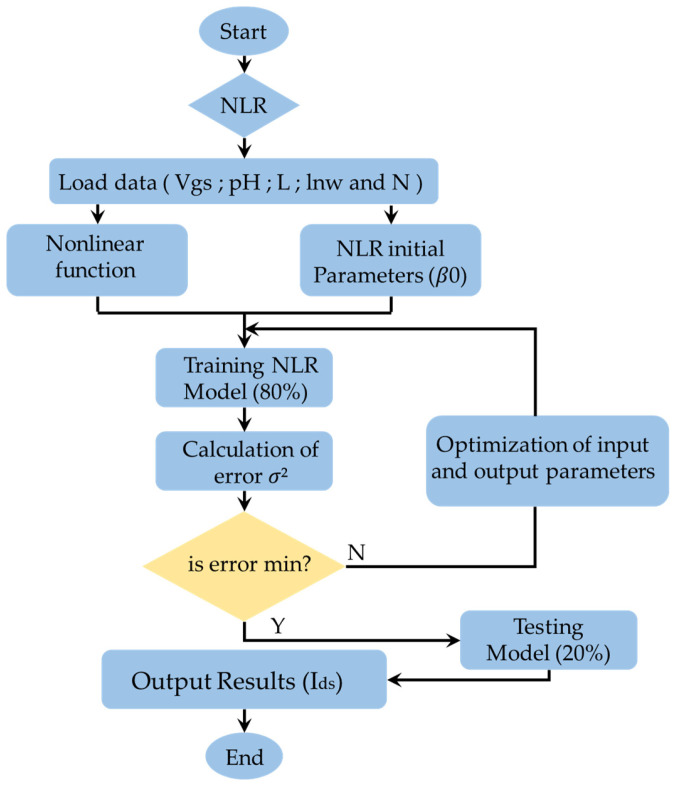
Program flow chart of the Nonlinear Regression (NLR).

**Figure 7 sensors-24-08091-f007:**
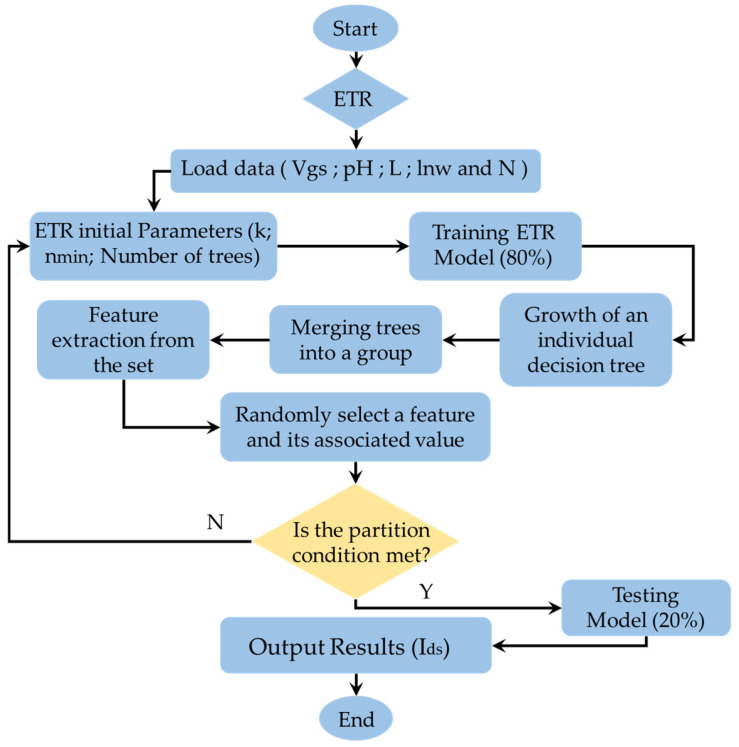
Program flow chart of the Extra Trees regression (ETR).

**Figure 8 sensors-24-08091-f008:**
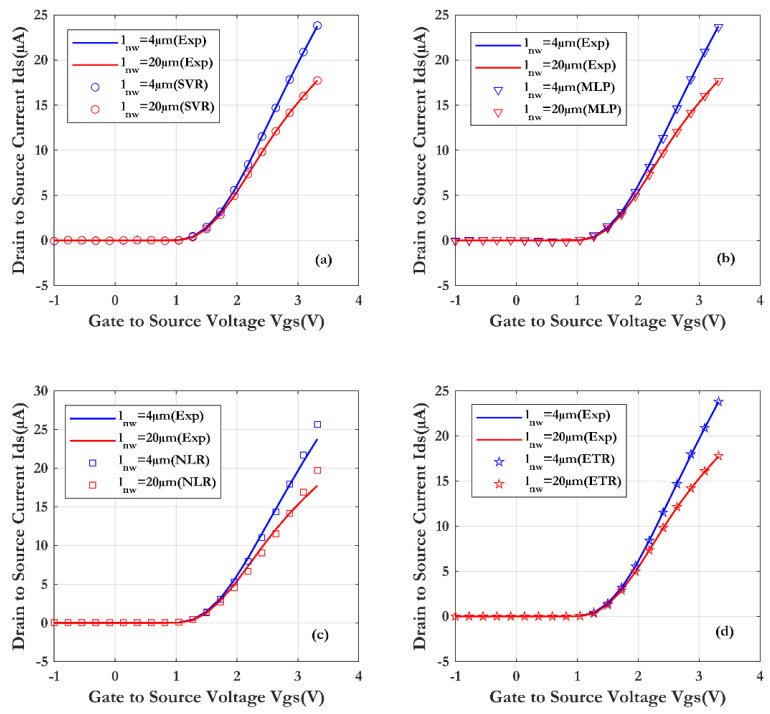
Comparison of SiNW-ISFET characteristics with varying wire lengths, obtained through four machine learning techniques and validated by measurements (Vds = 1 V): (**a**) SVR, (**b**) MLP, (**c**) NLR and (**d**) ETR.

**Figure 9 sensors-24-08091-f009:**
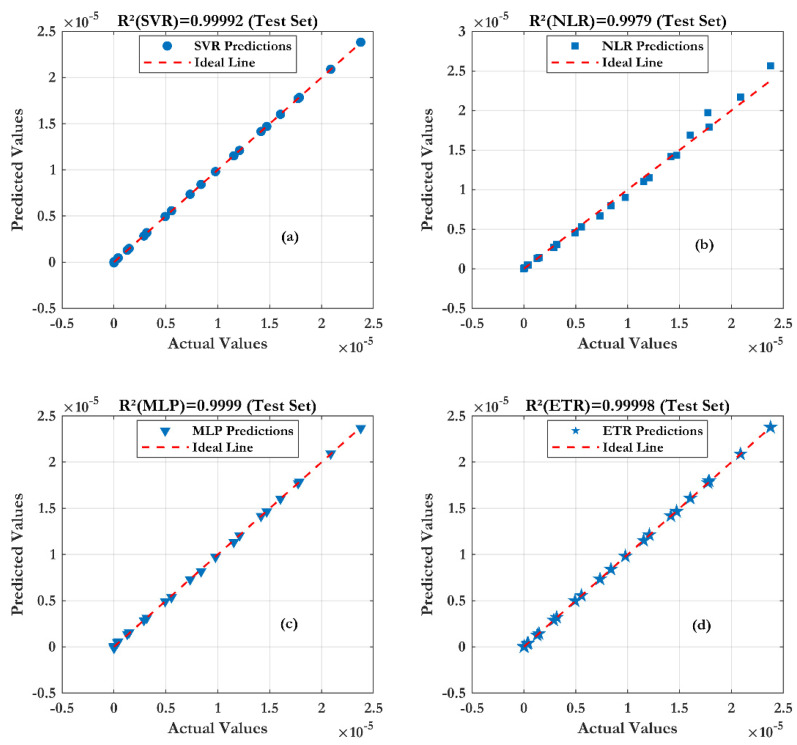
Prediction accuracy of four machine learning techniques with different lengths of nanowires: (**a**) SVR, (**b**) NLR, (**c**) MLP and (**d**) ETR.

**Figure 10 sensors-24-08091-f010:**
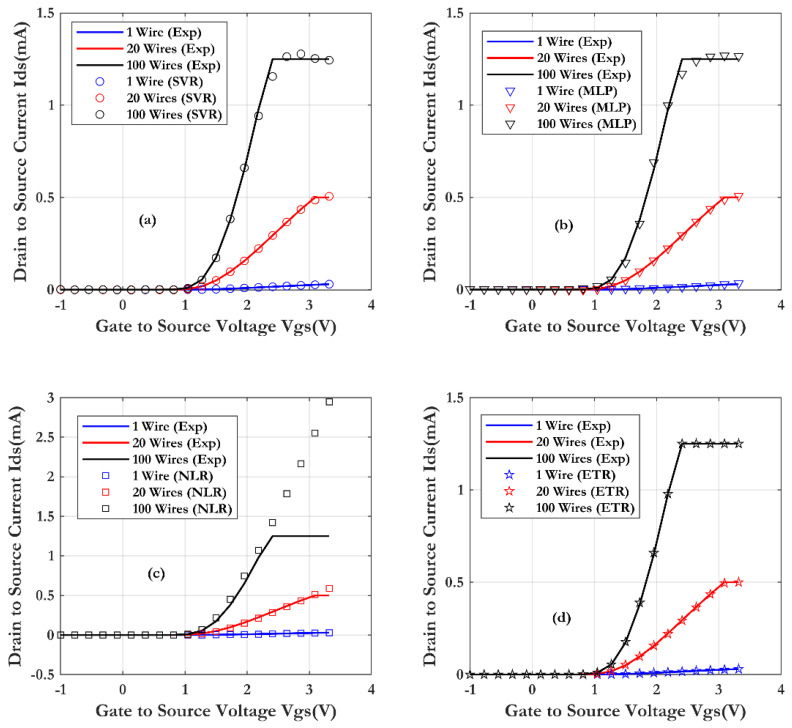
Comparison of the SiNW-ISFET characteristics with different numbers of wires, obtained through four machine learning techniques and validated by measurements (the nanowire length is 2 μm and Vds = 1 V): (**a**) SVR, (**b**) MLP, (**c**) NLR and (**d**) ETR.

**Figure 11 sensors-24-08091-f011:**
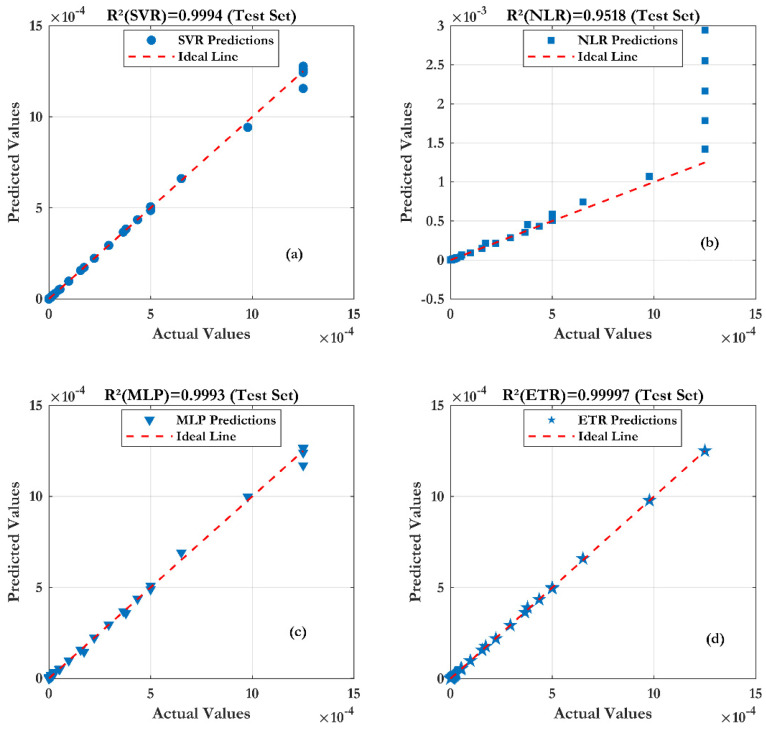
Prediction accuracy of four machine learning techniques with different numbers of wires: (**a**) SVR, (**b**) NLR, (**c**) MLP and (**d**) ETR.

**Figure 12 sensors-24-08091-f012:**
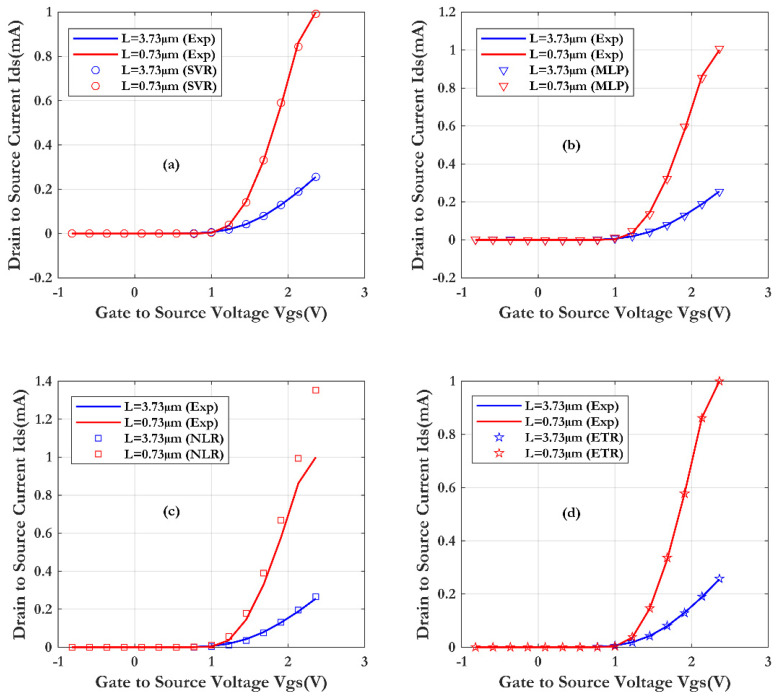
Comparison of the SiNW-ISFET characteristics with different gate lengths (0.73 μm and 3.73 μm), obtained through four machine learning techniques and validated by measurements (the nanowire length is 10 μm and Vds = 1 V): (**a**) SVR, (**b**) MLP, (**c**) NLR and (**d**) ETR.

**Figure 13 sensors-24-08091-f013:**
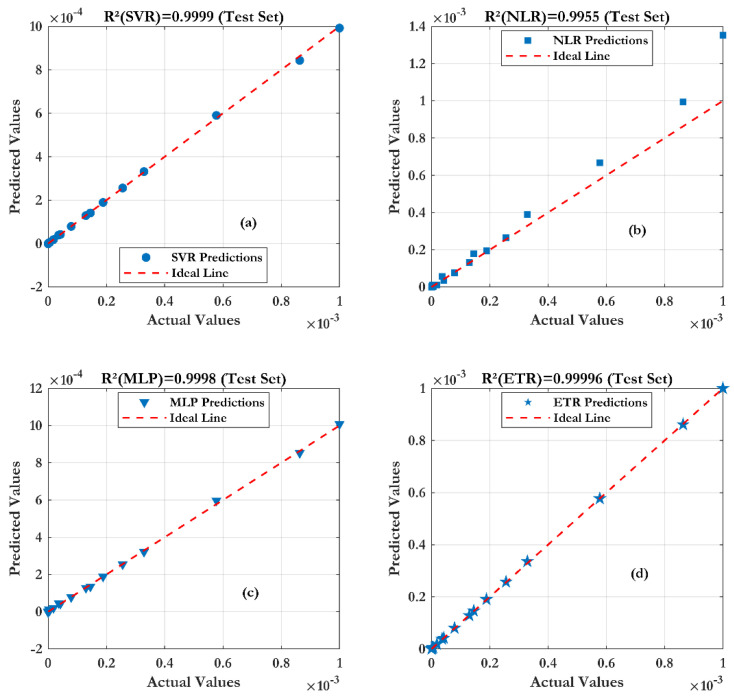
Prediction accuracy of four machine learning techniques with different gate lengths: (**a**) SVR, (**b**) NLR, (**c**) MLP and (**d**) ETR.

**Figure 14 sensors-24-08091-f014:**
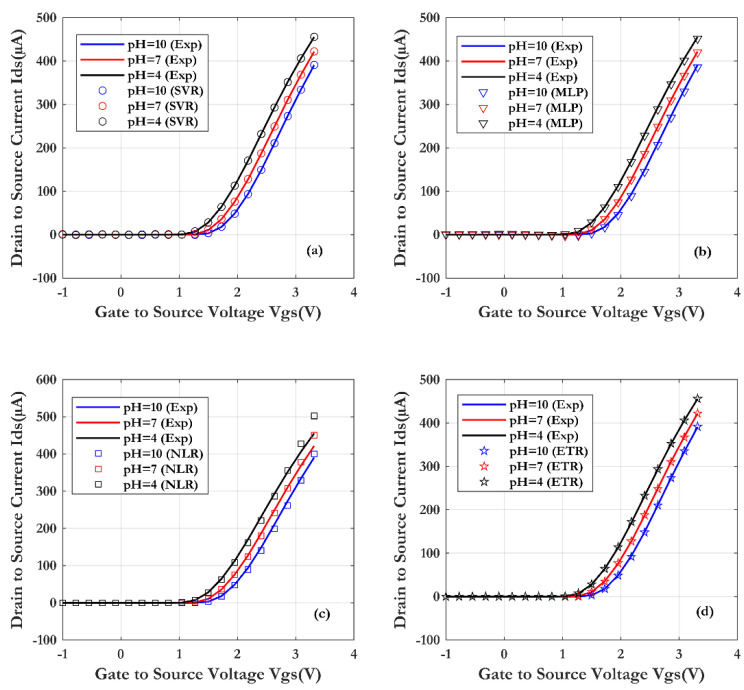
Comparison of the SiNW-ISFET characteristics with different pH, obtained through four machine learning techniques and validated by measurements (Vds = 1 V): (**a**) SVR, (**b**) MLP, (**c**) NLR and (**d**) ETR.

**Figure 15 sensors-24-08091-f015:**
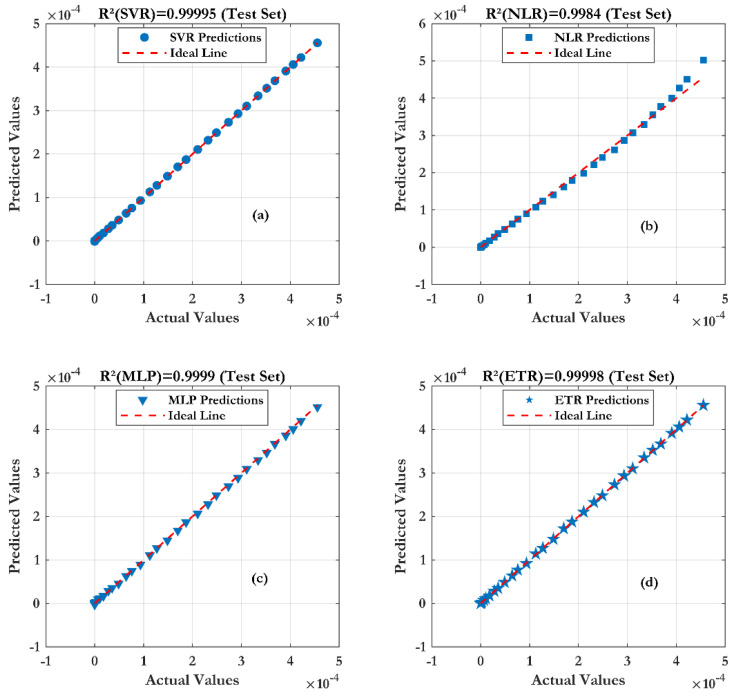
Prediction accuracy of four machine learning techniques with different pH values: (**a**) SVR, (**b**) NLR, (**c**) MLP and (**d**) ETR.

**Figure 16 sensors-24-08091-f016:**
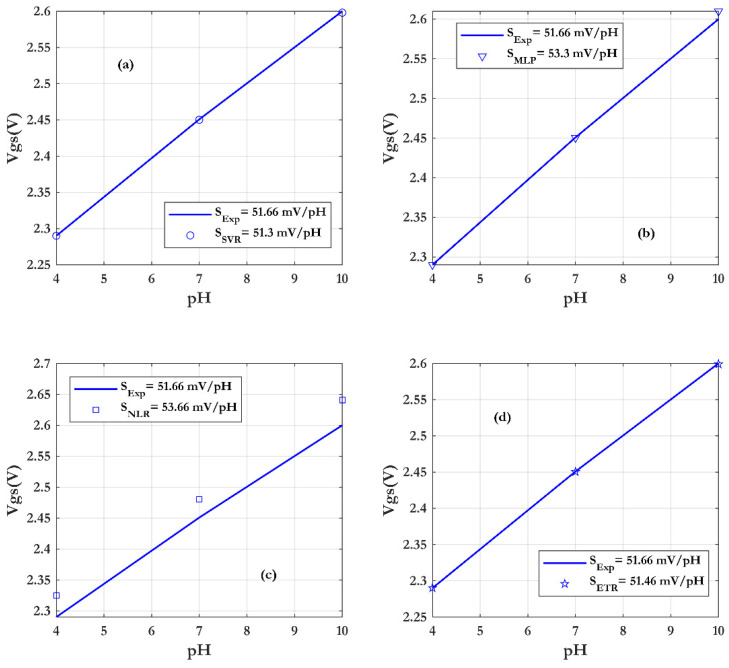
Comparison of the sensitivities of SiNW-ISFET sensor, obtained through four machine learning techniques and validated by measurements (Ids = 200 µA and Vds = 1 V): (**a**) SVR, (**b**) MLP, (**c**) NLR and (**d**) ETR.

**Table 1 sensors-24-08091-t001:** Error indicators.

Metric	Formula
$R^{2}$	$1-\frac{\sum_{i=1}^{n}{\left(y_{i}-{\hat{y}}_{i}\right)}^{2}}{\sum_{i=1}^{n}{\left(y_{i}-\bar{y}\right)}^{2}}$
MAE	$\frac{1}{n}\sum_{i=1}^{n}\left|y_{i}-{\hat{y}}_{i}\right|$
RMSE	$\sqrt{\frac{1}{n}\sum_{i=1}^{n}{\left(y_{i}-{\hat{y}}_{i}\right)}^{2}}$

where n represents the number of elements of data from the whole test process; $y_{i}$
and ${\hat{y}}_{i}$
are the correct and predicted values of target [47].

**Table 2 sensors-24-08091-t002:** Data ranges used for ML.

Range	Length of Nanowire (µm)	Number of Wires	Gate Length (µm)	Vgs (V)	pH	Ids (mA)
min	4	1	0.73	−1	4	10^−9^
max	20	100	3.73	3.5	10	1.25
		Input				Output

**Table 3 sensors-24-08091-t003:** Comparison the uncertainty indices obtained for the machine learning models.

	RMSE (mA)	MAE (mA)	R^2^ (%)
MLP	0.0053	0.00264	99.9725
NLR	0.095	0.028	98.59
SVR	0.0047	0.00156	99.97925
ETR	0.001	0.000415	99.99725

**Table 4 sensors-24-08091-t004:** Parameters selected for different machine learning techniques.

Model	Parameters
MLP	activation function (γl): “softmax”, alpha: 0.001, batch size: ‘auto’, number of hidden layer: 1, hidden layer size (s): 10, learning rate (α): 10^−8^, max-iter: 200
SVR	penalty applied over training errors (C): 10, epsilon: 10^−8^, gamma(γ): auto, kernel function (G):‘Gaussian’, max-iter: 1, termination tolerance (tol): 10^−4^
NLR	fit-intercept: True, termination tolerance for the function (TolFun): 10^−4^
ETR	Number of trees: 100, n_min_: 2, k: 1

## Data Availability

The original contributions presented in this study are included in the article. Further inquiries can be directed to the corresponding author.

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
