# Peer review of "Machine Learning-Based Modeling of pH-Sensitive Silicon Nanowire (SiNW) for Ion Sensitive Field Effect Transistor (ISFET)"

_sensors, 2024, doi:10.3390/s24248091_

Round 1
Reviewer 1 Report
Comments and Suggestions for Authors
Referee Report
On the paper “ Machine learning-based modelling of pH-sensitive silicon nanowire (SiNW) ion sensitive field effect transistor (ISFET) “ (sensors-3304687) by the authors Nabil Ayadi, Ahmet Lale, Bekkay Hajji, Jérôme Launay and Pierre Temple-Boyer submitted to the Sensors
This is interesting theoretical paper. It reports a model for the SiNW-based ISFET sensor aimed at predicting its performance and key parameters. Four innovative numerical models have been developed using different ML techniques: Multilayer Perceptron (MLP), Nonlinear Regression (NLR), Support Vector Regression (SVR), and Extra Trees Regression (ETR). The accuracy of these models was assessed by comparing simulation results with experimental data. From results, a good agreement has been noted. In addition, the comparison of the four models shows the effectiveness of the models based on the (ML) techniques. In particular, the Extra Trees Regression (ETR) allows optimal performances with a low error rate (RMSE of 1×10-3 mA and R2 of 0.9999725). The experimental data are reliable without any doubts. However, I have some questions about structure and magnetic performance of obtained samples. I would like to note a few points to improve the paper before it can be published:
1. Everything the motivation should be deleted from the Abstract.
2. The authors should provide in 1. Introduction examples of promising silicon based materials:
(1). R.F. Abreu, S.O. Saturno, F.A.C. Nobrega, D. da M. Colares, J.P.C. do Nascimento, S.J.T. Vasconcelos, F.E.A. Nogueira, D.B. de Freitas, F.F. do Carmo, A. Ghosh, T.O. Abreu, M.A.S. Silva, R.S. Silva, S.V. Trukhanov, D. Zhou, C. Singh, A.S.B. Sombra, Study of electrical properties with temperature variation by complex impedance spectroscopy (CIS) and efects on the Ba2TiSi2O8–TiO2 matrix, Appl. Phys. A 130 (2024) 138. https://doi.org/10.1007/s00339-024-07295-z.
3. I understand the choice of the object of study. This is the metallic alloys nanoparticles based samples which have excellent electronic properties. I fully agree with the authors that: “ To address these issues, the development of a new silicon nanowire (SiNW)based ISFET sensor seems to be the ideal candidate, offering a good sensitivity caused by high surface-to-volume ratio [11,12,13 and 14]. ”. However, the metal, hydroxide, and their alloys are not free from some disadvantages. One of which is their low resistance to the aggressive influence of environmental factors such as temperature, oxygen and electromagnetic radiation. The oxide compounds in this sense are much more stable when used up to 1000 ºC. There is such class of materials as the complex iron oxides with excellent electronic properties that are also promising for practical application:
(2). S.V. Trukhanov, A.V. Trukhanov, V.G. Kostishyn, L.V. Panina, An.V. Trukhanov, V.A. Turchenko, D.I. Tishkevich, E.L. Trukhanova, V.V. Oleynik, O.S. Yakovenko, L.Yu. Matzui, D.A. Vinnik, Magnetic, dielectric and microwave properties of the BaFe12-xGaxO19 (x ≤ 1.2) solid solutions at room temperature, J. Magn. Magn. Mater. 442 (2017) 300-310. https://doi.org/10.1016/j.jmmm.2017.06.022.
This information should be mentioned in 1. Introduction.
4. For oxides, metals, their alloys and composites the stoichiometry is particularly important. The deviation from stoichiometry and appearance of the oxygen anions can lead to a change in the charge state of the cations, which in turn will greatly change the electronical parameters. That will seriously affect the practical application of the materials obtained. What is the oxygen stoichiometry of prepared composite samples? It is well known that the complex transition metal compounds easily allow the oxygen excess and/or deficit:
(3). S.V. Trukhanov, A.V. Trukhanov, A.N. Vasil'ev, A. Maignan, H. Szymczak, Critical behavior of La0.825Sr0.175MnO2.912 anion-deficient manganite in the magnetic phase transition region, J. Exp. Theor. Phys. Lett. 85 (2007) 507-512. https://doi.org/10.1134/S0021364007100086.
Data on the oxygen stoichiometry of the obtained samples should be given in the paper and discussed in the context of their relationship with capacitance properties. This should be discussed in 3. Results and discussion.
5. It is necessary to discuss in more details how the average crystallite size of investigated samples and crystallite size distribution change with annealing temperature and how they affect their electrochemical properties. Mass transport, and especially oxygen transport, destroys the grains:
(4). S.V. Trukhanov, I.O. Troyanchuk, V.V. Fedotova, V.A. Ryzhov, A. Maignan, D. Flahaut, H. Szymczak, R. Szymczak, Magnetic properties of the nonstoichiometric Sr-doped manganites, Phys. Stat. Solidi (b) 242 (2005) 1123-1131. https://doi.org/10.1002/pssb.200402143.
This should be also discussed in 3. Results and discussion.
6. The presented 4 papers should be inserted in References.
The paper should be sent to me for the second analysis after the major revisions.
Author Response
Thank you very much for taking the time to review this manuscript. Please find the detailed responses below and the corresponding revisions/corrections highlighted/in track changes in the re-submitted files.
This is interesting theoretical paper. It reports a model for the SiNW-based ISFET sensor aimed at predicting its performance and key parameters. Four innovative numerical models have been developed using different ML techniques: Multilayer Perceptron (MLP), Nonlinear Regression (NLR), Support Vector Regression (SVR), and Extra Trees Regression (ETR). The accuracy of these models was assessed by comparing simulation results with experimental data. From results, a good agreement has been noted. In addition, the comparison of the four models shows the effectiveness of the models based on the (ML) techniques. In particular, the Extra Trees Regression (ETR) allows optimal performances with a low error rate (RMSE of 1×10-3 mA and R2 of 0.9999725). The experimental data are reliable without any doubts. However, I have some questions about structure and magnetic performance of obtained samples. I would like to note a few points to improve the paper before it can be published:
- Everything the motivation should be deleted from the Abstract.
Response 1: Thank you for your important comment.
The novelty has been clarified and the summary completely modified.
Modified summary:
The development of ion-sensitive field-effect transistor (ISFET) sensors based on silicon nanowires (SiNW) has recently seen significant progress, due to their many advantages such as compact size, low cost, robustness and real-time portability. However, little work has been done to predict the performance of SiNW-ISFET sensors. The present study focuses on predicting the performance of the silicon nanowire (SiNW)-based ISFET sensor using four machine learning techniques, namely multilayer perceptron (MLP), nonlinear regression (NLR), support vector regression (SVR) and extra tree regression (ETR). The proposed ML algorithms are trained and validated using experimental measurements of the SiNW-ISFET sensor. The results obtained show a better predictive ability of extra tree regression (ETR) compared to other techniques, with a low RMSE of 1×10-3 mA and an R2 value of 0.9999725. This prediction study corrects the problems associated with SiNW -ISFET sensors.
- The authors should provide in 1. Introduction examples of promising silicon based materials:
(1). R.F. Abreu, S.O. Saturno, F.A.C. Nobrega, D. da M. Colares, J.P.C. do Nascimento, S.J.T. Vasconcelos, F.E.A. Nogueira, D.B. de Freitas, F.F. do Carmo, A. Ghosh, T.O. Abreu, M.A.S. Silva, R.S. Silva, S.V. Trukhanov, D. Zhou, C. Singh, A.S.B. Sombra, Study of electrical properties with temperature variation by complex impedance spectroscopy (CIS) and efects on the Ba2TiSi2O8–TiO2 matrix, Appl. Phys. A 130 (2024) 138. https://doi.org/10.1007/s00339-024-07295-z.
Response 2: Thank you for your important comment. In the introduction, we've added examples of promising sensing nanomaterials such as SiWN and carbon nanotubes (CNT) used for ISFET chemical sensors. This clarification has been added to the new manuscript and the added text is as follows:
To overcome these problems, researchers are exploring the potential of nanomaterials and nanotechnologies to develop small, low-cost, highly efficient and sensitive biosensors with a wide range of applications. Silicon nanowires (SiNW) and carbon nanotubes (CNT) are examples of promising sensing materials. Their size, comparable to that of biological species, makes silicon nanowires (SiNW) an ideal interface between biological media and physicists' tools. What's more, their high surface-to-volume ratio makes them particularly sensitive to surface charge effects. These unique and unexpected properties of nanoscale materials make them ideal candidates for sensing applications.
The development of the ISFET sensor based on silicon nanowires (SiNW) appears to be the ideal candidate, offering good sensitivity thanks to a high surface-to-volume ratio [11,12,13 and 14]. What's more, SiNW-ISFET sensors are inexpensive, easy to manufacture and reliable.
- I understand the choice of the object of study. This is the metallic alloys nanoparticles based samples which have excellent electronic properties. I fully agree with the authors that: “ To address these issues, the development of a new silicon nanowire (SiNW)based ISFET sensor seems to be the ideal candidate, offering a good sensitivity caused by high surface-to-volume ratio [11,12,13 and 14]. ”. However, the metal, hydroxide, and their alloys are not free from some disadvantages. One of which is their low resistance to the aggressive influence of environmental factors such as temperature, oxygen and electromagnetic radiation. The oxide compounds in this sense are much more stable when used up to 1000 ºC. There is such class of materials as the complex iron oxides with excellent electronic properties that are also promising for practical application:
(2). S.V. Trukhanov, A.V. Trukhanov, V.G. Kostishyn, L.V. Panina, An.V. Trukhanov, V.A. Turchenko, D.I. Tishkevich, E.L. Trukhanova, V.V. Oleynik, O.S. Yakovenko, L.Yu. Matzui, D.A. Vinnik, Magnetic, dielectric and microwave properties of the BaFe12-xGaxO19 (x ≤ 1.2) solid solutions at room temperature, J. Magn. Magn. Mater. 442 (2017) 300-310. https://doi.org/10.1016/j.jmmm.2017.06.022.
This information should be mentioned in 1. Introduction.
Response 3: Thanks for the comments.Yes, this information is mentioned in the introduction.
Metallic alloy nanoparticles possess excellent electronic properties, but their stability is limited due to low resistance to environmental factors, such as temperature, oxygen, and electromagnetic radiation. In contrast, oxide compounds, including complex iron oxides, provide greater stability at temperatures up to 1000 ºC, making them promising for practical applications [47].
This clarification has been added to the new manuscript.
- For oxides, metals, their alloys and composites the stoichiometry is particularly important. The deviation from stoichiometry and appearance of the oxygen anions can lead to a change in the charge state of the cations, which in turn will greatly change the electronical parameters. That will seriously affect the practical application of the materials obtained. What is the oxygen stoichiometry of prepared composite samples? It is well known that the complex transition metal compounds easily allow the oxygen excess and/or deficit:
(3). S.V. Trukhanov, A.V. Trukhanov, A.N. Vasil'ev, A. Maignan, H. Szymczak, Critical behavior of La0.825Sr0.175MnO2.912 anion-deficient manganite in the magnetic phase transition region, J. Exp. Theor. Phys. Lett. 85 (2007) 507-512. https://doi.org/10.1134/S0021364007100086.
Data on the oxygen stoichiometry of the obtained samples should be given in the paper and discussed in the context of their relationship with capacitance properties. This should be discussed in 3. Results and discussion.
Response 4: Thanks for the comments.
The Al2O3 detection layer was deposited by ALD. The thickness of this layer is about 3.3nm for 30 ALD cycles, and about 4.5nm for 50 ALD cycles. The composition of this layer at the interface evolves very gradually from silicon to alumina Al2O3, we will call it aluminosilicate AlxSiyOz. The alumina beyond the interface zone is stoichiometric, after calibration of the EDX with two samples, one in sapphire Al2O3 and the other in SiO2, the ratio between oxygen and aluminum is indeed 60:40 [48].
This clarification has been added to the new manuscript.
- It is necessary to discuss in more details how the average crystallite size of investigated samples and crystallite size distribution change with annealing temperature and how they affect their electrochemical properties. Mass transport, and especially oxygen transport, destroys the grains:
(4). S.V. Trukhanov, I.O. Troyanchuk, V.V. Fedotova, V.A. Ryzhov, A. Maignan, D. Flahaut, H. Szymczak, R. Szymczak, Magnetic properties of the nonstoichiometric Sr-doped manganites, Phys. Stat. Solidi (b) 242 (2005) 1123-1131. https://doi.org/10.1002/pssb.200402143.
This should be also discussed in 3. Results and discussion.
Response 5: Thanks for the comments.
It has been shown that the refractive index of alumina deposited by ALD increases under the effect of heat treatment. This phenomenon could be explained by crystallisation of the films. To demonstrate this, the alumina was characterised using grazing incidence X-ray diffraction (GIXRD). This is a non-destructive technique used to study crystalline structures. The sample is subjected to an X-ray with a wavelength close to the dimensions of the crystal lattice (around 0.1 nm). The crystalline planes then act as a lattice of slits whose spacing is close to the wavelength of the X-rays. Diffraction then occurs at a precise angle that depends on the crystalline structure of the sample being analysed. If the structure is amorphous, there is no crystalline plane and therefore no diffraction [49].
This clarification has been added to the new manuscript.
- The presented 4 papers should be inserted in References.
Response 6: Thank you, the 4 suggested articles have been inserted into the references.

Reviewer 2 Report
Comments and Suggestions for Authors
1. On page 3, line 102, why is the output expressed as a vector when it’s just a single number? Wouldn't it be more straightforward to express it directly as a single value?
2. On page 3, line 106, what does "no enough training packages" mean? Are you referring to an insufficient number of software tools, such as frameworks, libraries, etc.? What programming language and frameworks did you use for model training?
3. On page 3, line 128, should "80%" and "20%" be followed by "of all the data"?
4. On page 3, line 131, is it reasonable for I_ds to equal y_j here? According to the definition in Problem 1, Y = (y_1, y_2, ..., y_n) represents current I_ds, so I_ds should be a vector of outputs. However, later it’s used as an element of the output vector.
5. Between lines 149 and 150 on page 4, is "b_ik" defined? (I couldn’t find it in the article.)
6. Between lines 155 and 156 on page 5, is there a missing right parenthesis in Equation (10)? The number of left and right parentheses seems mismatched.
7. On page 5, line 158, the backpropagation proof is provided only for the second layer. Should the proof for the first layer (i.e., w_1jk) also be included, or could relevant papers on backpropagation be referenced and this proof omitted?
8. For Figure 3, should the dataset be specified and the algorithm defined for each iteration? Should the output arrow from the “N” decision box connect to the “Define the learning algorithm” box, as shown? Is the 80% dataset derived from shuffling the full dataset for each iteration? If so, there’s a risk that the training set might contain elements from the test set, potentially causing data leakage and model performance bias.
9. On page 6, lines 171-172, n is defined as the dimension of the vector, but both x and y use the same subscript i. Must the dimensions of x and y vectors (i.e., input and output vector dimensions) be the same? This also appears on page 8, line 221.
10. In Equation (15) on page 6, Equation (16), and lines 201 and 205 on page 7, are the subscripts x_i and y_i correctly marked?
Comments on the Quality of English LanguageN.A.
Author Response
Thank you very much for taking the time to review this manuscript. Please find the detailed responses below and the corresponding revisions/corrections highlighted/in track changes in the re-submitted files.
- On page 3, line 102, why is the output expressed as a vector when it’s just a single number? Wouldn't it be more straightforward to express it directly as a single value?
Response 1: Thank you for pointing out the error in equation 1. We have expressed the output Y, which is the current Ids, directly as a single value X in the new corrected manuscript.
- On page 3, line 106, what does "no enough training packages" mean? Are you referring to an insufficient number of software tools, such as frameworks, libraries, etc.? What programming language and frameworks did you use for model training?
Response 2: Thank you for your insightful remark. All algorithms used in this work are implemented using popular machine learning libraries in Python, such as scikit-learn and TensorFlow. These libraries offer a wide range of tools and features to efficiently build, train, and evaluate machine learning models (pandas, numpy, matplotlib, etc.).
Based on 956 experimental data (for each experimental Ids-Vgs curve), 80% of the data (i.e. 765 data) were used to train the machine learning models and the remaining 20% ​​(i.e. 191 data) were used to test the machine learning models.
All these corrections have been made to the new manuscript.
- On page 3, line 128, should "80%" and "20%" be followed by "of all the data"?
Response 3: Thank you for this good question.
A split ratio of 80%–20% is used to divide the data into training and testing sets. Based on 956 experimental data (for each experimental Ids-Vgs curve), 80% of the data (i.e. 765 data) were used to train the machine learning models and the remaining 20% ​​(i.e. 191 data) were used to test the machine learning models.
This clarification has been added to the new manuscript.
- On page 3, line 131, is it reasonable for I_ds to equal y_j here? According to the definition in Problem 1, Y = (y_1, y_2, ..., y_n) represents current I_ds, so I_ds should be a vector of outputs. However, later it’s used as an element of the output vector.
Response 4: Thank you for your question. Y = (y_1, y_2, ..., y_n) does represent the current I_ds, which is a vector of outputs. The relationship was correctly corrected in question 2.
This is simply a typing error.
- Between lines 149 and 150 on page 4, is "b_ik" defined? (I couldn’t find it in the article.)
Response 5: Thank you for your question. The coefficient bl expressed in equation 5 was defined on page 141 as the bias term.
- Between lines 155 and 156 on page 5, is there a missing right parenthesis in Equation (10)? The number of left and right parentheses seems mismatched.
Response 6: Very good point. Equation (10) is missing a right-hand parenthesis. This parenthesis has been added and the number of left and right parentheses in equation 10 is now correct.
- On page 5, line 158, the backpropagation proof is provided only for the second layer. Should the proof for the first layer (i.e., w_1jk) also be included, or could relevant papers on backpropagation be referenced and this proof omitted?
Response 7: Thank you for your important question. Indeed, the proof of backpropagation is provided only for the second layer, while the proof for the first layer (i.e. w_1jk) can be omitted by referring to established work on backpropagation. A relevant reference on this subject is :
Popescu, M. C., Balas, V. E., Perescu-Popescu, L., & Mastorakis, N. (2009). Multilayer perceptron and neural networks. WSEAS Transactions on Circuits and Systems, 8(7), 579-588.
- For Figure 3, should the dataset be specified and the algorithm defined for each iteration? Should the output arrow from the “N” decision box connect to the “Define the learning algorithm” box, as shown? Is the 80% dataset derived from shuffling the full dataset for each iteration? If so, there’s a risk that the training set might contain elements from the test set, potentially causing data leakage and model performance bias.
Response 8: Thank you for your valuable comments.
For Figure 3, should the dataset be specified and the algorithm defined for each iteration?
Iteration is applied only to the training dataset; it is not necessary to specify the dataset or redefine the algorithm for each iteration in Figure 3.
Should the output arrow from the “N” decision box connect to the “Define the learning algorithm” box, as shown?
This is an error in the arrow which has been corrected in figure 3..
Is the 80% dataset derived from shuffling the full dataset for each iteration?
No, the 80% dataset is not derived from shuffling the full dataset for each iteration.
Figure 3 illustrates the functioning of the Multilayer Perceptron (MLP) algorithm. The process begins with downloading the dataset, followed by preparing the input and output variables. Next, the input features are normalized to ensure a uniform scale across the dataset. The model is then defined and trained on 80% of the data, using the backpropagation gradient algorithm for training. Evaluation is based on calculating the error e (the difference between Ids calculated and Ids measured), which should be minimized to assess the model’s performance across multiple subsets of the dataset. Once trained, the models are deployed to predict new data instances. Finally, the error indicators R2, RMSE, and MAE are used to effectively evaluate the predictive capability of the model.
This clarification has been added to the new manuscript.
- On page 6, lines 171-172, n is defined as the dimension of the vector, but both x and y use the same subscript i. Must the dimensions of x and y vectors (i.e., input and output vector dimensions) be the same? This also appears on page 8, line 221.
Response 9: Thank you for your question. In lines 171-172, n is defined as the dimension of the vector, and although both x and y share the same subscript i, their dimensions (i.e., input and output vector dimensions) do necessarily need to be identical. This also applies to page 8, line 221. This concept is further illustrated in the following article: LT-FS-ID: Log-Transformed Feature Learning and Feature-Scaling-Based Machine Learning Algorithms to Predict the k-Barriers for Intrusion Detection Using Wireless Sensor Networks.
- In Equation (15) on page 6, Equation (16), and lines 201 and 205 on page 7, are the subscripts x_i and y_i correctly marked?
Response 10: Thank you for your question; Yes, the subscripts xi and xj have been reviewed and are correctly marked in Equation (15) on page 6, as well as in Equation (16) and on lines 201 and 205 on page 7. This is further demonstrated in the following article: LT-FS-ID: Log-Transformed Feature Learning and Feature-Scaling-Based Machine Learning Algorithms to Predict the k-Barriers for Intrusion Detection Using Wireless Sensor Networks.

Round 2
Reviewer 1 Report
Comments and Suggestions for Authors
Referee Report
On the paper “ Machine learning-based modelling of pH-sensitive silicon nanowire (SiNW) ion sensitive field effect transistor (ISFET) “ (sensors-3304687-v2) by the authors Nabil Ayadi, Ahmet Lale, Bekkay Hajji, Jérôme Launay and Pierre Temple-Boyer submitted to the Sensors
This paper has been well corrected and it can be recommended.
